# Inferring 3D Occupancy Fields through Implicit Reasoning on Silhouette Images

## ABSTRACT

Implicit 3D representations have shown great promise in deep learning-based 3D reconstruction. With differentiable renderers, current methods are able to learn implicit occupancy fields without 3D supervision by minimizing the error between the images rendered from the learned occupancy fields and 2D ground truth images. In this paper, however, we hypothesize that a full rendering pipeline including visibility determination and evaluation of a shading model is not required for the learning of 3D shapes without 3D supervision. Instead, we propose to use implicit reasoning, that is, we reason directly on the implicit occupancy field without explicit rendering. This leads our method to reveal highly accurate 3D structures from low quality silhouette images. Our implicit reasoning infers a 3D occupancy field by evaluating how well it matches with multiple 2D occupancy maps, using occupancy clues rather than rendering the 3D occupancy field into images. We exploit the occupancy clues that indicate whether a viewing ray inside a 2D object silhouette hits at least one occupied 3D location, or whether a ray outside the silhouette hits no occupied location. In contrast to differentiable renderers whose losses do not distinguish between the inside and outside of objects, our novel loss function weights unoccupied clues more than occupied ones. Our results outperform recent state-of-the-art techniques, justifying that we can learn accurate occupancy fields only using sparse clues without an explicit rendering process.

## CCS CONCEPTS

• **Do Not Use This Code → Generate the Correct Terms for Your Paper**; *Generate the Correct Terms for Your Paper*; Generate the Correct Terms for Your Paper; Generate the Correct Terms for Your Paper.

## KEYWORDS

Neural rendering, Differentiable renderer, Silhouette Images, Neural implicit function, Occupancy.

**Unpublished working draft. Not for distribution.**

*ACM MM, 2024, Melbourne, Australia*
© 2024 Copyright held by the owner/author(s). Publication rights licensed to ACM.
ACM ISBN 978-x-xxxx-xxxx-x/YY/MM
https://doi.org/10.1145/nnnnnnn.nnnnnnn

## 1 INTRODUCTION

Implicit functions have emerged as an important 3D representation in deep learning models. They enable deep neural networks to represent 3D shapes in a discriminative manner by learning mappings from 3D locations to their occupancy labels (or signed distance values). For learning with 3D supervision, 3D locations with known occupancy labels are sampled densely around 3D ground truth shapes, which are used as training samples.

More recent methods [5, 8, 12, 20, 21, 27, 29, 37–39, 41, 46, 47, 49] proposed various differentiable renderers to implement surface rendering or volume rendering , which enable the learning of implicit functions from 2D images without 3D supervision. These differentiable renderers first render implicit functions learned by deep neural networks into 2D images, and then refine the implicit function by back-propagating the error between the rendered and ground truth images. Although differentiable renderers provide an intuitive way to optimize the learned implicit functions to match the ground truth shapes, this kind of explicit reasoning requires implementing a complete rendering procedure including visibility testing (via rasterization or ray tracing) or even shading in a differentiable manner. This also makes some of these methods rely on high quality images as supervision which provides perfect multi-view color consistency for structure inference. In contrast, in this paper we hypothesize that we can still conduct the learning of deep implicit shape representations without a complete rendering pipeline.

To investigate this question, we propose implicit reasoning to learn 3D occupancy fields for 3D shapes from multiple 2D occupancy maps, that is, low quality silhouette images. Without explicit rendering, our implicit reasoning infers a 3D occupancy field by evaluating how well it matches with multiple silhouette images. Here, "implicit" means that we reason only based on the implicit occupancy function, but we do not explicitly evaluate visibility or shading. Specifically, implicit reasoning leverages two types of occupancy clues that we evaluate on the 2D image plane. The first one is the occupied clue, that is, if we shoot a ray from a pixel inside a 2D silhouette to the 3D shape, there must be at least one occupied 3D location along the ray. The second one is the unoccupied clue, that is, if we shoot a ray from a pixel outside a 2D silhouette to the 3D shape, there cannot be any occupied 3D location along the ray. To conduct implicit reasoning, we introduce a loss function for each ray that includes two terms to evaluate how well the learned implicit function fits these two clues. Our approach leads to two main insights, which are first that the unoccupied clues are more important than the occupied clues, and second, clues from a sparse set of pixels are adequate to conduct good implicit reasoning on the

implicit function. Finally, we report state-of-the-art results that justify our approach in the experiments. In summary, our main contributions are as follows:

i) We propose implicit reasoning to infer 3D occupancy fields from multiple silhouette images without rendering. Our novel loss function can implicitly evaluate how well the learned 3D implicit function matches the unknown ground truth.

ii) In contrast to rendering based methods that infer 3D structures from different pixels in the same way, we justified the feasibility of inferring 3D structures in different manners determined by 2D occupancy labels, which provides a novel perspective to more efficiently leverage 2D supervision.

iii) We demonstrate that our method improves the accuracy over the state-of-the-art in 3D reconstruction from single images under widely used benchmarks.

## 2  RELATED WORK

In recent years, deep learning based 3D structure learning has made huge progress with supervised or unsupervised methodology [10, 31–33, 42, 43]. Because of the limit of pages, we will only briefly review methods trained without 3D supervision.

**For explicit 3D representations.** We have been able to leverage deep learning models to learn from different explicit 3D representations including voxel grids [6, 35, 36, 45], triangle meshes [2, 13, 17–19], point clouds [11, 14, 15, 26, 48]. Without 3D supervision, a widely used strategy is to propose differentiable renderers to first render a reconstructed 3D shape into 2D images, and then, calculate the error between the rendered and ground truth images, where the error on 2D images can be back propagated to train the neural networks through the differentiable renders.

For voxel grids, differentiable renderers were proposed to bridge 3D voxels to silhouette images by projective projection with shooting rays [45] or orthogonal projection with simple projection function [6]. For the projection, the camera positions can be known [6, 45], estimated by an additional network along with the reconstruction network [35] or in the presence of viewpoint uncertainties [6]. To model the information along each ray, [45] selected the maximum occupancy value along a ray and [36] derived a differentiable formulation by leveraging ray collision probabilities.

For triangle meshes, OpenDR [22] was first proposed to approximate gradients with respect to pixel positions in back-propagation. With hand-crafted gradients [13] or analytically compute gradients [17, 18], the loss on 2D images can also be back-propagated to update vertices on meshes. Similarly, SoftRas [19] introduced a probabilistic rasterization to assign each pixel to all faces on a mesh, and Chen et al. [2] employed interpolation of local mesh properties as the rasterization.

Different from voxel grids and meshes, point clouds can not directly represent a 3D continuous surface. To resolve this issue, different renderers employed either dense points [15] or different rasterization [11, 14, 26, 48] in rendering process.

For example, Lin et al. [15] proposed pseudo-renderer to model the visibility using pooling among the projections of dense 3D points. Rendering based methods employed surface splatting [48] or Gaussian functions in 3D space [11] and on 2D images [14, 26] to approximate the point distribution on point clouds.

**For implicit 3D representations.** Recently, it has been drawing more research interests to learn implicit 3D representations using deep learning models. With 3D supervision, implicit 3D representations provide deep learning models a way to generate 3D shapes in a discriminative manner [3, 23, 24, 28, 30, 33, 40], such as classifying a 3D location into inside or outside of a 3D shape or regressing its signed distance value. Without 3D supervision, different differentiable renderers employed rendering process to learn implicit 3D representations by back propagating loss on 2D images.

Specifically, to reduce the computational cost on sampling implicit surface required in training, Vincent et al. [34] proposed differentiable ray marching to learn a mapping from world coordinates to a feature representation of local scene properties. To render a signed distance field, [21] introduced another differentiable sphere tracing method which runs efficiently with affordable memory consumption. With indifferentiable sphere tracing, SDFDiff [12] was proposed with a differentiable shading to back propagate the loss on 2D images. With the analytically computed normal, a method [50] introduced the rendering of SDF by projecting 0-isosurface with surface tangent discs. To render an occupancy field, Liu et al. [20] employed ray-based field probing technique to probe the learned implicit function, and rendered silhouette images by aggregating occupancy along each ray using max pooling. Similarly, Wu et al. [44] rendered occupancy field by aggregating detection points on rays to mine supervision for 3D occupancy fields. Using the concept of implicit differentiation, depth gradients were derived analytically in a differentiable rendering formulation for implicit shape and texture representations [27]. By leveraging single views from objects as supervision, Lin et al. [16] managed to learn signed distance fields using a novel differentiable rendering formulation.

NeRF [25] was introduced to model geometry and appearance of shapes and scenes for novel view synthesis using volume rendering. Based on NeRF, recent methods [5, 29, 37, 39, 41, 46, 47, 49] learn implicit functions from multiview images with additional priors or constraints. However, they require high quality images for multi-view consistency to infer accurate 3D structures. While we merely use low quality silhouette images to reveal 3D structures and learn to reconstruct a shape from a single image.

All these differentiable renderers were proposed to explicitly infer the implicit 3D representation based on rendering. Our method is different from them by introducing implicit reasoning for implicit 3D representation without rendering.

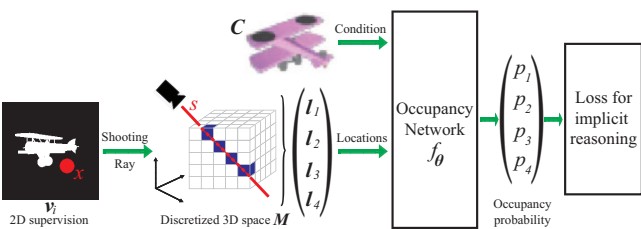

**Figure 1: Demonstration of implicit reasoning along rays.**

## 3 METHOD

### 3.1 Problem Statement and Overview

Without 3D supervision, we aim to learn implicit occupancy fields for 3D shapes using a neural network from multiple silhouette images $\{v_i | i \in [1, V]\}$. The neural network with a set of parameters $\theta$ represents the implicit occupancy field by learning a mapping $f_\theta$ from an observation condition $C$, such as an image, and a 3D location $l \in \mathbb{R}^3$ to the probability of occupancy $p$:

$$f_\theta : C \times l \rightarrow p, \text{where } p \in [0, 1]. \quad (1)$$

Without 3D supervision, current methods leverage differentiable renderers to produce rendered images from the learned implicit occupancy fields, which are then refined by minimizing the difference between the rendered images and the ground truth images. Our implicit reasoning accomplishes the same task, but without the needs of rendering an image with visibility testing (via ray tracing or rasterization) and evaluating some shading model. Instead, we introduce a novel loss only based on the implicit occupancy function to evaluate how well the occupancy probabilities $p$ fit the ground truth silhouettes.

We demonstrate our framework in Fig. 1. Our implicit reasoning is based on shooting rays $s$ from pixels $x$ on the 2D ground truth silhouette images $v_i$ to the space holding 3D shapes. Our occupancy function $f_\theta$ represents 3D shapes in an object-centered coordinate system, and we evaluate the occupancy function in a discretized 3D space $M$ with a resolution of $R$. Note that we uniformly discretize the 3D space $M$ to facilitate rays to probe the space rather than representing 3D shapes using voxel grids, since it would be easier to determine the coordinate of which 3D area is involved on each ray in implicit reasoning optimization. Moreover, our preliminary results show that we can achieve much better training efficiency than randomly sampling points along each ray with much less sampled points to train and much faster convergence, which still leads to our implicit reasoning to infer smooth occupancy fields. The discrete space not only reduce the redundance of sampled points but also improve the limited probing ability of randomly sampled points on rays.

For each ray $s$, we input the coordinates $l_j$ of the 3D grid cells hit by the ray to the occupancy network $f_\theta$, where $j \in [1, J]$ and $J$ is the number of grid cells hit by the ray.

Accordingly, the occupancy network predicts the occupancy probabilities for each $l_j$ with an observation condition $C$ as follows,

$$p_j = f_\theta(C, l_j). \quad (2)$$

Finally, our loss for implicit reasoning evaluates how the current occupancy probabilities $p_j$ fit the ground truth silhouettes. Then, the parameters $\theta$ in the occupancy network are further optimized to minimize the loss. For a simplified demonstration in Fig. 1, we only represent the 3D space $M$ with a resolution of $R = 5$, and shoot a ray hitting $J = 4$ 3D locations on $M$.

### 3.2 Clues on Silhouette Images

We conduct implicit reasoning solely based on occupancy clues provided by the 2D ground truth silhouette images $v_i$. As our experiments suggest, these clues are adequate to infer accurate implicit occupancy fields during training. We leverage two types of occupancy clues, including *occupied clues* and *unoccupied clues*:

i) Occupied clues: if a ray $s$ starts from a pixel $x$ inside a silhouette with a pixel value of $y = 1$, then there must be at least one occupied grid location among all $J$ cells traversed by the ray.

ii) Unoccupied clues: if a ray $s$ starts from a pixel $x$ outside a silhouette with a pixel value of $y = 0$, then there cannot be any occupied cells among the $J$ cells hit by the ray.

Note that we do not directly leverage the voxel grid to infer 3D structure, we only want to obtain coordinate to index 3D area. The grid will help us easily to obtain the coordinate of 3D area affected by a ray, and get better probing ability than using points directly sampled on rays.

We further illustrate these two kinds of clues in Fig. 2. For simplicity, we show the 3D space $M$ as a 2D grid and the 2D silhouette image $v_i$ as a 1D vector. The ray for the occupied clue starts from a pixel with a value of $y = 1$ and hits at least one occupied cell. While the ray for the unoccupied clue starts from a pixel with a value of $y = 0$ and does not hit any occupied cells.

### 3.3 Loss for Implicit Reasoning

A key of our approach is a novel loss to evaluate how well the currently learned implicit occupancy field fits with the occupied and unoccupied clues. This loss implements our implicit reasoning, which is based only on the implicit occupancy function but does not require explicit rendering. Minimizing the loss effectively refines the implicit occupancy fields as we show in our experiments.

For each ray $s$ starting from a pixel $x$ with a value of $y$, we first collect the 3D locations of the grid cells $l_j$ traversed by the ray $s$, where $j \in [1, J]$. Given a condition $C$, we get their occupancy probabilities $p_j$ by evaluating $p_j = f_\theta(C, l_j)$, as shown in Fig. 1.

For occupied clues, the challenge is that we cannot define a loss to specifically regulate how many occupied grid cells there should be along the ray, since the clues only require the

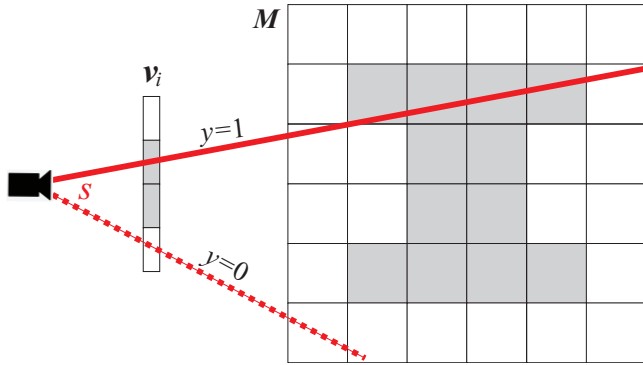

Figure 2: The occupied (solid line) and unoccupied (dotted line) clues. If the ray $s$ shoots from the silhouette on image $v_i$, it will hit at least one occupied area in the space of $M$, otherwise it hits no occupied area.

occupied number to be larger than one. However, the actual number of occupied cells is highly variable among different shapes. One option would be to use max-pooling to first select the cell with maximum occupancy probability, which is then further optimized to be as close to one as possible, as used in [20]. However, this leads to inefficient optimization because only one occupancy probability can be updated in the back propagation. To resolve this issue, we define a loss $O$ to aggregate all occupancy probabilities $p_j$ on each ray into an occupancy summary $A$ by summing them. Then, we normalize $O$ to lie in a range of $[0, 1]$ using an exponential function,

$$O = y \times exp(-A), \qquad (3)$$

where $A = \sum_{j=1}^{J} p_j$ and $y$ is the silhouette value of the pixel where the ray $s$ starts. Here, we loosen the occupied clue ($y = 1$) a little bit. $O$ is not definite zero when $A > 1$, such as $A \in [1, 5]$, although $O$ approaches zero along with increasing $A$. This is because it is highly possible for $A$ to be larger than one in the optimization when we sum up lots of small occupancy probabilities on the ray, although these small occupancy probability locations are not expected to be occupied. In addition, our loss defined for unoccupied clues with a weight can help to make up the loosen.

For unoccupied clues, our loss $U$ directly requires the occupancy summary $A$ along the ray to be as close to zero as possible. To normalize $U$ in a range of $[0, 1]$ similar as the loss $O$, we use the average of $A$ in $U$ as follows, where $J$ is the number of grid cells hit by the ray,

$$U = (1 - y) \times A/J. \qquad (4)$$

In summary, our loss function to conduct implicit reasoning is a weighted sum of $O$ and $U$,

$$E = O + \beta \times U, \qquad (5)$$

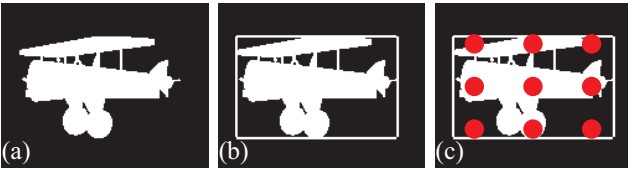

Figure 3: Illustration of shooting rays. We shoot rays within the bounding box of silhouette (b) on image (a) with a subsampling factor in (c).

where $\beta$ is a weight to balance the contribution from the occupied and unoccupied clues. Therefore, our objective function is to find an optimal set of parameters $\boldsymbol{\theta}$ in the occupancy network.

$$\boldsymbol{\theta}^* = \arg\min_{\boldsymbol{\theta}} O + \beta \times U. \qquad (6)$$

The contribution of our loss here is three-fold. First, our loss removes the requirement of rendering for the learning of implicit occupancy fields without 3D supervision. Instead, the loss only uses occupancy clues. Second, our loss allows us to infer accurate implicit occupancy fields with sparse rays from each view. Third, different from current rendering based methods that treat all pixels equally in the per-pixel loss computation, we find that we can increase the inference performance by weighting the loss $U$ for unoccupied clues more. More detailed implementation can be found in the code released in our supplementary material. We will justify our contributions in the following experiments.

## 4 EXPERIMENTS AND ANALYSIS

### 4.1 Details

**Dataset and metric.** For fair comparison, we leverage the widely used 13 categories from ShapeNetCore [1]. Specifically, in the auto-decoding experiments, we use 5 categories including airplane, car, chair, rifle, and table with the same train/test splitting as [3], where the resolution of shapes is $R = 64$. In the single image reconstruction experiments, we use all 13 categories with rendered images, ground truth voxel grids in a resolution of $R = 32$, and train/test splitting from [4]. In all experiments, we employ volumetric IoU to evaluate the accuracy of the reconstructed shapes. Note that all reported IoU values are multiplied by $10^2$.

**2D supervision and rays.** We only use $V = 20$ silhouette images as supervision for each 3D shape for network training. On each silhouette image, we merely shoot rays $s$ from sparse pixel locations to the 3D space with the known camera poses. Specifically, we first calculate the bounding box on a silhouette image, as shown in Fig. 3 (a) to (b), and then shoot rays from a subsampled set of pixels with a subsampling factor of 5, as illustrated by red dots in Fig. 3 (c). We found that this setting can help us save computational burden without degenerating the learning performance. We also explore the effect of subsampling factor in the following.

| Classes | Supervision | Plane | Bench | Cabinet | Car | Chair | Display | Lamp | Speaker | Rifle | Sofa | Table | Phone | Boat | Mean |
|---------|-------------|-------|-------|---------|-----|-------|---------|------|---------|-------|------|-------|-------|------|------|
| DISN [40] | Occupancy | 61.7 | 54.2 | 53.1 | 77.0 | 54.9 | 57.7 | 39.7 | 55.9 | **68.0** | 67.1 | 48.9 | 73.6 | 60.2 | 59.4 |
| DIBR [2] | RGB | 57.0 | 49.8 | 76.3 | 78.8 | 52.7 | 58.8 | 40.3 | 72.6 | 56.1 | 67.7 | 50.8 | 74.3 | 60.9 | 61.2 |
| SDFDiff [12] | | 68.7 | **68.6** | 77.4 | 80.0 | **64.4** | 65.8 | 51.5 | 65.3 | 55.5 | **76.5** | **62.9** | 82.8 | 62.4 | 66.7 |
| PTNR [45] | Silhouette | 55.6 | 48.8 | 57.1 | 65.2 | 35.1 | 39.6 | 29.1 | 46.0 | 51.3 | 53.1 | 31.0 | 67.0 | 40.8 | 47.7 |
| PTN [45] | | 55.6 | 49.2 | 68.2 | 71.2 | 44.9 | 54.0 | 42.2 | 58.7 | 59.9 | 62.2 | 49.4 | 75.0 | 55.1 | 57.4 |
| NMR [13] | | 58.5 | 45.7 | 74.1 | 71.3 | 41.4 | 55.5 | 36.7 | 67.4 | 55.7 | 60.2 | 39.1 | 76.2 | 59.4 | 57.0 |
| SoftRas [19] | | 58.4 | 44.9 | 73.6 | 77.1 | 49.7 | 54.7 | 39.1 | 68.4 | 62.0 | 63.6 | 45.3 | 75.5 | 58.9 | 59.3 |
| IMFun [44] | | 53.3 | 39.1 | 65.2 | 66.0 | 44.4 | 52.2 | 37.7 | 62.7 | 38.9 | 54.8 | 45.6 | 66.8 | 53.8 | 52.3 |
| IMRender [20] | | 65.1 | 53.6 | - | 78.2 | 54.8 | - | - | - | - | - | 51.5 | - | 60.8 | 60.7 |
| Ours | Silhouette | **73.1** | 62.4 | **77.6** | **86.7** | 61.8 | **69.5** | **52.7** | 76.9 | 62.9 | 69.7 | 61.0 | **84.6** | **68.1** | **69.8** |

**Table 1: Quantitative comparison of single image 3D shape reconstruction in terms of IoU.**

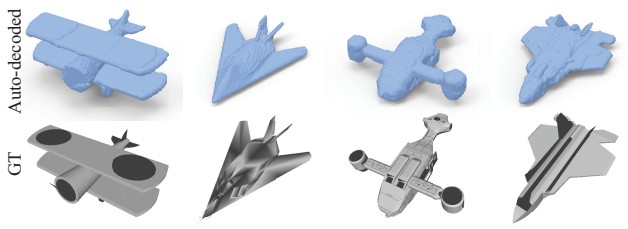

**Figure 4: Auto-decoded shapes.**

**Network and training.** The occupancy network $f_{\boldsymbol{\theta}}$ that we are using is modified from OccNet [23] by replacing the 2D CNN encoder which extracts 512-dimensional features of observation $\boldsymbol{C}$. Specifically, in auto-decoding, we replace the OccNet 2D CNN encoder by a learnable feature for each 3D object, which is similar as shape memories [9] or codes [30]. In single image reconstruction, we replace the OccNet 2D CNN encoder by the 2D CNN network in SoftRas [19], which aims to better fit the 2D observation $\boldsymbol{C}$ in the dataset from [4] in unsupervised learning application.

We train our network using the Adam optimizer with a learning rate of 0.0001 and a batch size of 400 rays. In each epoch, we iterate over all shapes in the training set in random order. In each batch, all 400 rays are randomly shot from all $V = 20$ views of the same 3D shape, with the same condition $\boldsymbol{C}$ that is also randomly selected from the images rendered for the same 3D shape (from dataset [4] for single image reconstruction). Along each ray, we randomly repeat some hit voxels to make all rays containing the same number of hit voxels. We initialize the balance weight $\beta$ in our loss in Eq. 5 as 30.

During testing, for each 3D shape, we first employ the learnable features (in auto-decoding) or 2D observation $\boldsymbol{C}$ (in single image reconstruction) as condition to generate occupancy values at all voxels in the discretized 3D space $\boldsymbol{M}$ under a specific resolution of $R$. Then, we use marching cubes to obtain meshes without any further post-processing.

## 4.2 Auto-decoding

We first evaluate the occupancy network $f_{\boldsymbol{\theta}}$ trained by our loss in auto-decoding. We aim to reveal the structure of a 3D shape from a given set of silhouette images. To make

the implicit occupancy fields fit the silhouette images, we minimize our loss in Eq. 5 by optimizing the learnable features of shapes and the parameters $\boldsymbol{\theta}$ in the occupancy network at the same time. We show some complicated shapes auto-decoded by the learned occupancy network with a resolution of $R = 128$ in Fig. 4. The results demonstrate that our implicit reasoning can infer accurate 3D structures on shapes with arbitrary topology, such as the airplanes with layered or thin wings, which justifies the effectiveness of our loss.

We further visualize the optimization of two shapes in auto-decoding experiment in Fig. 5. We select several optimization steps to visualize each shape from 4 view angles. We found that our method can gradually infer the correct 3D structures using the clues on multiple silhouette images.

## 4.3 Single Image Reconstruction

We further evaluate the occupancy network trained by our loss in single image reconstruction, where we aim to reconstruct a 3D shape from a given rendered image. Following the setting of [20], we compare with the state-of-the-art methods trained without 3D supervision under 13 classes in Table 1 with a resolution of $R = 32$. Several recent methods mainly leverage silhouette images as 2D supervision, including perspective transform net [45] (PTN), the retrieval version of PTN (PTNR), neural mesh renderer [13] (NMR), soft rasterizer [19] (SoftRas), Implicit Function renderer (IMFun) [44] and implicit renderer [20] (IMRender). Other methods leverage RGB images as 2D supervision, such as interpolation-based differentiable renderer [2] (DIBR) and differentiable renderer for signed distance fields [12] (SDFDiff). Moreover, these differentiable renderers are designed for various 3D representations, such as voxel grids [45], triangle meshes [2, 13], signed distance fields [12], and implicit occupancy fields [20]. We achieve the best performance among the methods using silhouette images as 2D supervision, and significantly better than rendering-based methods [20, 44] for implicit occupancy fields. Our performance is a little bit worse than SDFDiff under the Bench, Chair, and Table classes. One possible reason is that SDFDiff leverages normals to calculate shading when rendering signed distance fields to compare with RGB images as the supervision signal. However, to evaluate shading, S-DFDiff requires that the illumination and surface reflectance

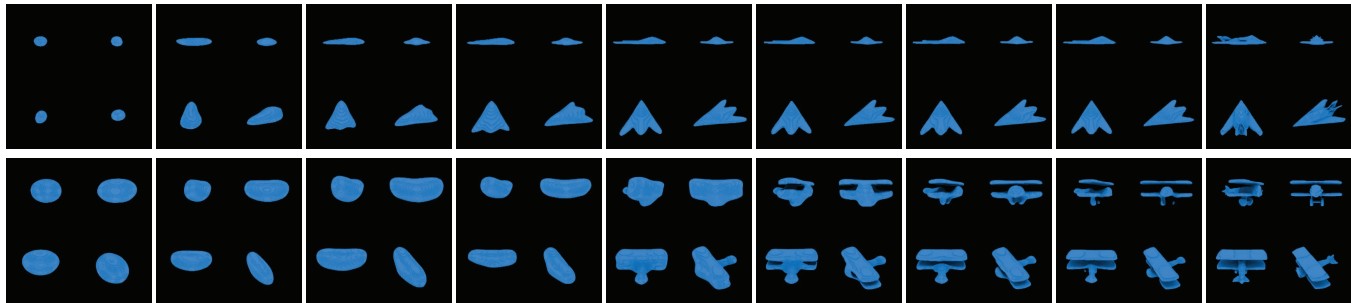

**Figure 5: Visualization of optimization in auto-decoding. The two shapes are visualized from four views at different optimization steps. Please watch our video in supplementary materials for more details.**

model is known. In contrast, our method does not require any such information.

Moreover, we also report the comparison with DISN [40] which leverages the 3D supervision to learn. We can see that our method significantly outperforms DISN under almost all classes. One thing we want to note is that although DISN can leverage 3D supervision to train the neural network, there are lots of training queries for each shape, which makes the network to learn the mapping from single images and 3D shapes.

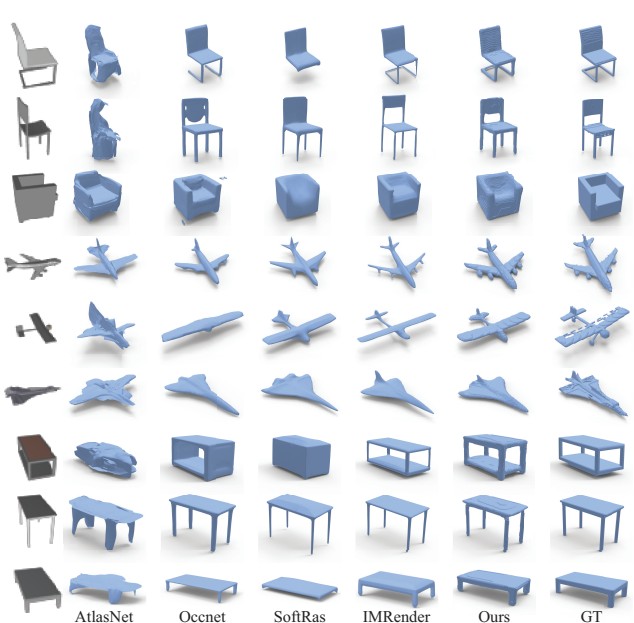

**Figure 6: Visual comparison with methods (Atlas-Net [7], OccNet [23]) trained with 3D supervision and methods (SoftRas [19], IMRender [20]) trained without 3D supervision under chair, plane and table classes.**

We further compare our method with a resolution of $R = 128$ with methods trained with and without 3D supervision under three challenging classes including chair, plane,

| Subsampling Factor | 1 | 2 | 4 | 5 | 7 |
|---|---|---|---|---|---|
| IoU | 71.1 | 71.5 | 72.3 | **73.1** | 71.3 |

**Table 2: Effect of number of rays in terms of IoU (%).**

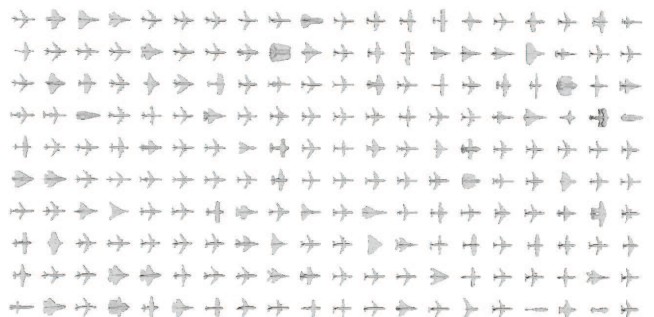

**Figure 7: The randomly selected 200 airplane reconstruction.**

table in Fig. 6. We can see that our method can reconstruct shapes with arbitrary topologies. In addition, compared to the GT shapes, our method reconstructs shapes with more accurate geometry than supervised methods and rendering-based unsupervised methods. Note that we also visualize the GT shapes in a resolution of $R = 128$ to highlight the similarity of our method to the GT shapes. However, with additional geometric regularization, IMRender [20] can produce smoother surfaces than ours. We further highlight our advantage by visualizing more single image reconstruction results obtained by our method in Fig. 7, Fig. 8, and Fig. 9, where we randomly select 200 from each one of airplane, chair and table class from the test dataset.

## 4.4 Ablation Study

**Sparse Rays.** One advantage of removing rendering is that we do not have to calculate information at every pixel location on each supervision plane. Therefore, our loss can

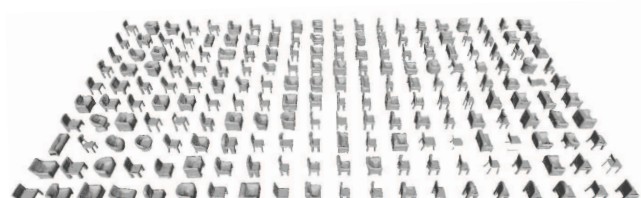

**Figure 8: The randomly selected 200 chair reconstruction.**

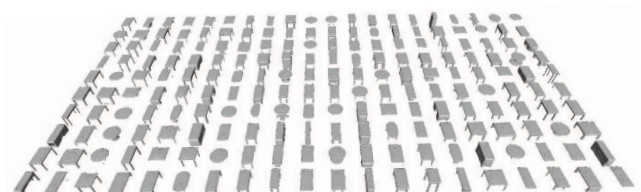

**Figure 9: The randomly selected 200 table reconstruction.**

| $\beta$ | 0 | 1(no O) | 10 | 15 | 20 | 25 | 30 | 35 |
|---|---|---|---|---|---|---|---|---|
| IoU | 15.58 | 0 | 61.9 | 68.3 | 71.3 | 71.4 | **73.1** | 71.2 |

**Table 3: Effect of weights in terms of IoU (%).**

leverage sparse rays in a multiple view scenario, as described in Fig. 3, to reveal plausible 3D structures. In this experiment, we explore the effect of the number of rays on the performance under the Airplane class in single image reconstruction. Shooting rays in the bounding box as shown in Fig. 3, we evaluate different subsampling factor in the loss calculation such as $\{1, 2, 4, 5, 7\}$. The IoU in Table 2 shows that the performance is not significantly affected by the number of rays even for a subsampling factor of 7.

**Weight.** Intuitively, rendering based methods also employ the occupied clues and unoccupied clues, but encoding them into rendered pixel values with equal weight. Our loss for implicit reasoning shows that it is helpful to increase the performance by weighting unoccupied clues more in the learning process, such as our $\beta = 30$ in our previous experiments. In this experiment, we explore the effect of the weight $\beta$ on the performance under the airplane class in single image reconstruction by using values in $\{0, 1(noO), 10, 15, 20, 25, 30, 35\}$. Our results in Table 3 show that the performance is increasing along with increasing $\beta$. But we can not get reasonable results when only using occupied clues ("0") or unoccupied clues ("1(no O)"). For example, when we only use the occupied clues, the generated shapes can not reveal 3D structures like shapes, while the network learns unoccupied everywhere when only using unoccupied clues.

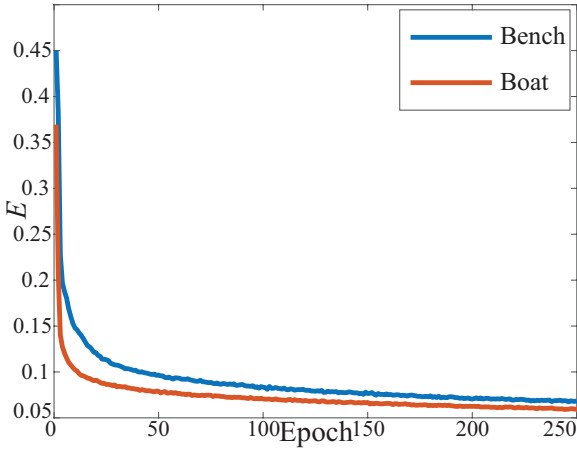

**Figure 10: Our loss under Bench and Boat during training.**

| View Number $V$ | 5 | 10 | 15 | 20 | 25 |
|---|---|---|---|---|---|
| IoU | 45.6 | 52.5 | 65.4 | **73.1** | **73.1** |

**Table 4: Effect of view numbers in terms of IoU (%).**

**Resolution.** In this experiment, we explore the effect of our loss on probing the discretized 3D space $\boldsymbol{M}$ with a resolution of $R$ under the airplane class. We keep the balance weight $\beta = 30$ and shoot sparse rays with subsampling factor of 5. We train the occupancy network $f_\theta$ with different resolutions, such as $\{32, 64, 128, 256\}$. The reconstruction comparison in Fig. 11 shows that our implicit reasoning is able to infer shapes at arbitrary resolutions with sparse rays. Note that we need to shoot denser rays in high resolution such as $R = 256$ to make up the degeneration caused by less locations hit by the rays, such as the improvement by rays with a subsampling factor of 2 in Fig. 11 (e) over a subsampling factor of 5 in Fig. 11 (d).

**View Number.** We also explore the effect of view numbers $V$. We reported our results with $V = 20$ in tables above. In this experiment, we try different different view number options to train the network, including $V = \{5, 10, 15, 20, 25\}$. We train the network using shapes in the training set under plane class, and evaluate the trained network using shapes in the testing set under the same class. We report our results in Tab. 4. The numerical comparison shows that our method can not inference the 3D structure from fewer views, and we can not get significant improvements if we use more than 20 views for each shape.

**Interpolation.** We visualize the learned global feature space by shape interpolation in Fig. 12. We randomly select two reconstructed shapes in the test set, and employ their latent codes to interpolate several new latent codes between them. The interpolated latent codes are further used to generate

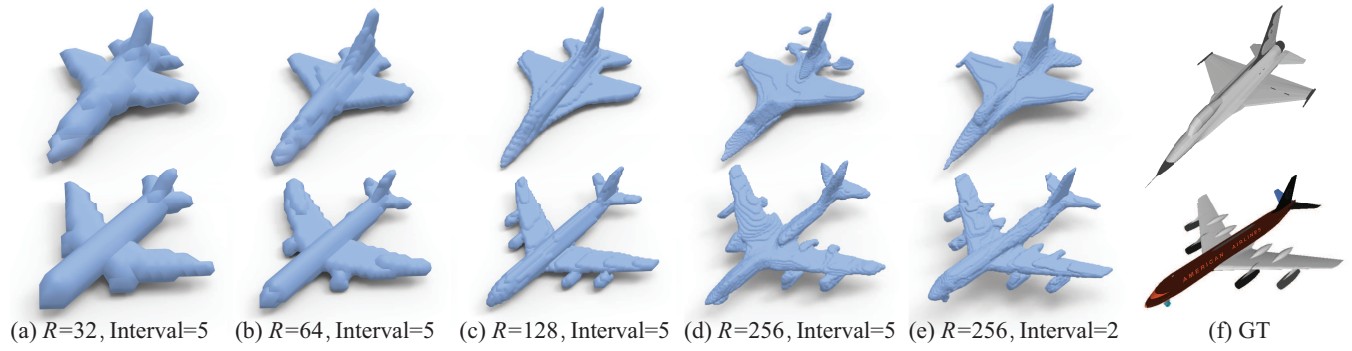

(a) $R$=32, Interval=5    (b) $R$=64, Interval=5    (c) $R$=128, Interval=5    (d) $R$=256, Interval=5    (e) $R$=256, Interval=2      (f) GT

**Figure 11: Shapes reconstructed by our method at different resolutions $R$.**

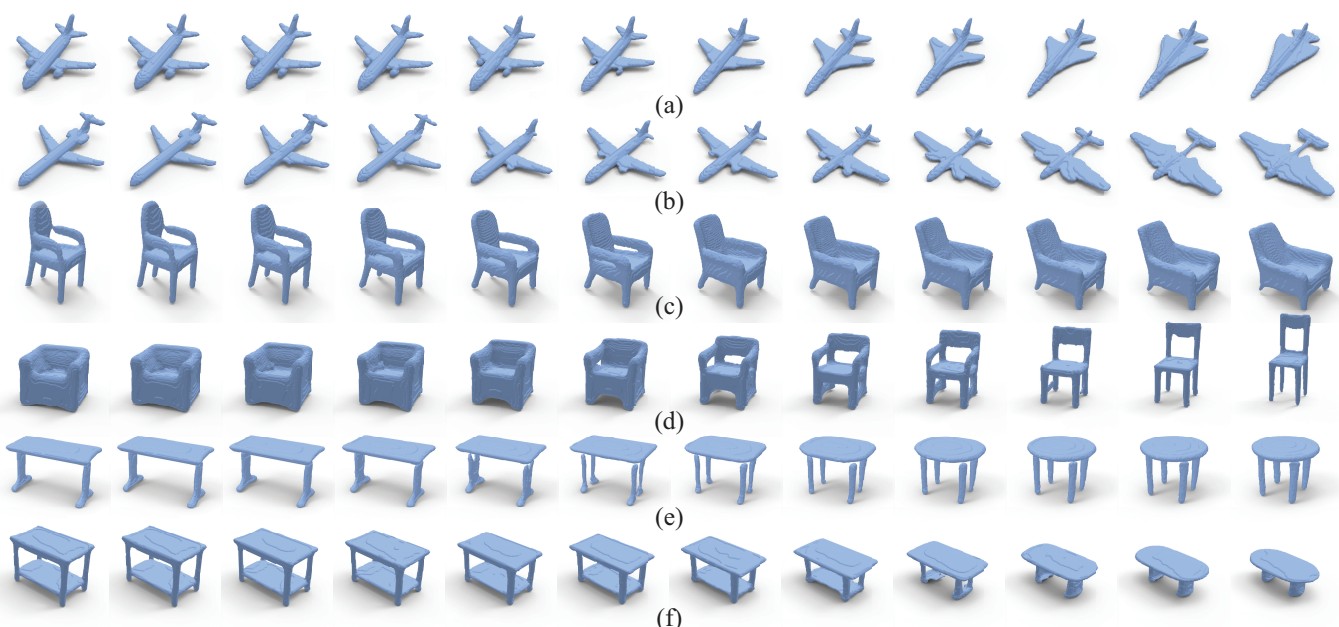

**Figure 12: Illustration of interpolated planes in (a-b), chairs in (c-d), and tables in (e-f).**

novel shapes by the trained network. We visualize two pairs of shape interpolation under each one of Airplane, Table, and Chair class. The interpolated shapes can smoothly transform from one shape to another, which justifies the semantics of the space learned with our loss.

**Loss.** Finally, we visualize our proposed loss using the loss curves producing the results of bench and boat in Fig. 10. We can see that our loss can smoothly approach to zero, which is also the approximate value of our loss computed on the GT shapes. The loss curves show that our proposed loss function can guide the occupancy network $f_\theta$ to converge very fast in the first several epochs under both classes.

## 5 CONCLUSION

We introduce implicit reasoning on silhouette images to infer implicit occupancy fields without rendering. Our implicit reasoning successfully leverages our proposed loss to evaluate how well the currently learned implicit occupancy fields fit the occupied and unoccupied clues on silhouette images. With only sparse clues, our performance shows significant improvement over rendering based methods. Different from RGB images, our renderer is able to infer geometry information from low quality supervision. More importantly, we show that it improves the performance if we weight differently on pixel supervision during reasoning. Our method justifies the feasibility of learning implicit occupancy fields without rendering, and provides a novel perspective to infer 3D structures in different manners determined by 2D occupancy labels, which leverages 2D supervision more efficiently than rendering based methods that adopt the same inference for all 2D occupancy labels.

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
