# OpenReview forum: "Inferring 3D Occupancy Fields through Implicit Reasoning on Silhouette Images"
_acmmm.org/ACMMM/2024/Conference — MM2024 Poster_

### Official Review · Reviewer_Jbz3 · 2024-05-22

**Rating:** 4
**Confidence:** 2

**Summary:**

This paper presents a novel method for 3D reconstruction from silhouette images without relying on a full rendering pipeline, unlike previous methods that require extensive rendering, including visibility determination and shading evaluation. Instead, the authors propose an "implicit reasoning" technique, focusing directly on the implicit occupancy field. They introduce a new loss function that better discriminates between occupied and unoccupied regions in a 3D space, enhancing the learning accuracy from sparse and low-quality silhouette data. The paper demonstrates their method in terms of accuracy, using only occupancy clues instead of explicit rendering.

**Strengths:**

1. The method's ability to infer 3D occupancy fields directly from sparse and low-quality 2D silhouette images is a significant advantage. This direct use of 2D data without the need for dense sampling or high-quality imaging reduces the dependency on extensive pre-processing or data enhancement.

2. The introduction of the loss function that differentiates more effectively between occupied and unoccupied clues allows for more precise learning of the 3D structures is make sense. This loss function is key to the method's ability to accurately reconstruct 3D shapes from sparse data inputs.

2. I think the method has scalability and adaptability. For the reason that its reliance on simple occupancy clues rather than detailed textural or color information makes it adaptable to a variety of contexts and scalable to different levels of complexity in 3D shape reconstruction tasks.

**Limitations:**

1. There is a lack of real-world testing. The paper primarily discusses results based on synthetic or controlled datasets. Real-world applicability might differ and would benefit from testing in diverse, practical scenarios.
2. Dependence on Silhouette Quality: Although the method is robust to low-quality images, the degree to which silhouette quality can vary and still produce accurate results is not extensively discussed.

**Suitability:**

3

---

### Official Review · Reviewer_Pfqv · 2024-05-22

**Rating:** 5
**Confidence:** 2

**Summary:**

This paper works on 3D reconstruction from simple 2D supervisions, i.e. 2D silhouette images. The main claim of this paper is that the 3D occupancy field can be learned from multiple silhouette images without rendering. This paper presents a simplified learning pipeline with weighted losses on occupied and unoccupied clues. Various experiment results demonstrate the effectiveness of the proposed method.

**Strengths:**

1.	This paper presents a simple yet effective method to learn 3D shapes from weak supervisions (i.e. 2D silhouette images).
2.	The proposed weighted loss (Eq5) insightfully tackles problems of previous methods.
3.	Various experiment results and ablation studies to support the effectiveness of the proposed method.
4.	The paper is overall well-written and easy to follow.

**Limitations:**

1.	Lines 385-393 needs to be rephrased for better understanding.
2.	Network architecture (Line 490): (1) Reason for this specific design. (2) is the performance affected by the network architecture?
3.	Line 635: (1) Why conduct the qualitative experiments of R=128? (2) Are the models also retrained with R=128?
4.	Line 858: it would be better to discuss about other designs here, e.g. [20] (Lines 371-377)
5.	(minor) (1) Lines 37-40 need to be updated. (2) The order of the figures does not fully align with the order of the texts.

**Suitability:**

2

---

### Official Review · Reviewer_dANi · 2024-05-26

**Rating:** 3
**Confidence:** 3

**Summary:**

This paper presents a novel approach to 3D shape reconstruction using implicit reasoning on silhouette images, aiming to overcome the limitations of existing rendering-based methods. The authors propose a method that directly reasons on implicit occupancy fields without the need for explicit rendering, hypothesizing that a full rendering pipeline is not necessary for learning 3D shapes without 3D supervision.

**Strengths:**

1.	The paper addresses a critical need in computer vision and graphics by providing a method for 3D shape reconstruction from sparse silhouette images.
2.	The paper's motivation is clear and compelling.
3.	The paper is well-written and structured, effectively conveying the problem statement, proposed method, experimental results, and conclusions.

**Limitations:**

1. The paper proposes the use of implicit reasoning to infer 3D occupancy fields without explicit rendering. However, the explanation of how this implicit reasoning is achieved lacks clarity. The method relies on evaluating how well the occupancy probabilities fit ground truth silhouettes without explicitly defining the reasoning process. This ambiguity makes it difficult to understand the exact mechanism behind the proposed approach and raises questions about its robustness and reliability.
2. The paper proposes a novel loss function for implicit reasoning, but the rationale behind its design is not adequately justified. For instance, the loss function aggregates occupancy probabilities along rays without considering the spatial arrangement or geometric relationships between the occupied cells. This oversimplified approach may lead to suboptimal results and fails to capture the complexity of 3D scene geometry.
3. The paper primarily focuses on inferring 3D occupancy fields from static silhouette images, but it does not adequately address the challenges posed by dynamic scenes or moving objects.
4. The paper claims that its method outperforms state-of-the-art techniques in 3D reconstruction from single images, but the experimental evidence provided is insufficient to support this assertion.
5. The issue of dataset adequacy is indeed a significant concern.

**Suitability:**

3

---

### Meta-Review · Area_Chair_325N · 2024-06-27

**Recommendation:** Accept (Poster)
**Confidence:** 4

**Metareview:**

The paper initially received 1 Weak Accept, 1 Borderline Accept, and 1 Borderline Reject. After the rebuttal, one reviewer raised the score and the final ratings are 2 Weak Accept and 1 Borderline Reject.

Overall, the paper presents an intriguing method for learning 3D shapes from 2D silhouette images. Although there are some remaining concerns pointed out by Reviewer dANi, the AC believes that its contributions outweighed these shortcomings, many of which can/should be addressed in the future work.